

# Prediction and analysis of novel key genes ITGAX, LAPTM5, SERPINE1 in clear cell renal cell carcinoma through bioinformatics analysis

Yingli Sui, Kun Lu and Lin Fu

Institute of Chronic Disease, School of Basic Medicine, Qingdao University, Qingdao, Shandong, China

## ABSTRACT

**Background**. Clear Cell Renal Cell Carcinoma (CCRCC) is the most aggressive subtype of Renal Cell Carcinoma (RCC) with high metastasis and recurrence rates. This study aims to find new potential key genes of CCRCC.

**Methods**. Four gene expression profiles (GSE12606, GSE53000, GSE68417, and GSE66272) were downloaded from the Gene Expression Omnibus (GEO) database. The TCGA KIRC data was downloaded from The Cancer Genome Atlas (TCGA). Using GEO2R, the differentially expressed genes (DEG) in CCRCC tissues and normal samples were analyzed. Gene ontology (GO) and Kyoto Encyclopedia of Genes and Genomes (KEGG) enrichment analyses were performed in DAVID database. A protein-protein interaction (PPI) network was constructed and the hub gene was predicted by STRING and Cytoscape. GEPIA and Kaplan-Meier plotter databases were used for further screening of Key genes. Expression verification and survival analysis of key genes were performed using TCGA database, GEPIA database, and Kaplan-Meier plotter. Receiver operating characteristic (ROC) curve was used to analyze the diagnostic value of key genes in CCRCC, which is plotted by R software based on TCGA database. UALCAN database was used to analyze the relationship between key genes and clinical pathology in CCRCC and the methylation level of the promoter of key genes in CCRCC.

**Results**. A total of 289 up-regulated and 449 down-regulated genes were identified based on GSE12606, GSE53000, GSE68417, and GSE66272 profiles in CCRCC. The upregulated DEGs were mainly enriched with protein binding and PI3K-Akt signaling pathway, whereas down-regulated genes were enriched with the integral component of the membrane and metabolic pathways. Next, the top 35 genes were screened out from the PPI network according to Degree, and three new key genes ITGAX, LAPTM5 and SERPINE1 were further screened out through survival and prognosis analysis. Further results showed that the ITGAX, LAPTM5, and SERPINE1 levels in CCRCC tumor tissues were significantly higher than those in normal tissues and were associated with poor prognosis. ROC curve shows that ITGAX, LAPTM5, and SERPINE1 have good diagnostic value with good specificity and sensitivity. The promoter methylation levels of ITGAX, LAPTM5 and SERPINE1 in CCRCC tumor tissues were significantly lower than those in normal tissues. We also found that key genes were associated with clinical pathology in CCRCC.

**Conclusion**. ITGAX, LAPTM5, and SERPINE1 were identified as novel key candidate genes that could be used as prognostic biomarkers and potential therapeutic targets for CCRCC.

Corresponding author
Lin Fu, fulin@qdu.edu.cn

## INTRODUCTION

Kidney cancer is a complex disease composed of a variety of cancers, showing different histology, clinical course, genetic changes, and response to treatment (*Linehan et al., 2019*). Renal Cell Carcinoma is the most common tumor in the kidney, Whose morbidity and mortality are rising worldwide. Renal cell carcinoma is divided into different subtypes, including clear cell renal cell carcinoma (CCRCC), chromogenic cell renal carcinoma (chRCC), and papillary renal cell carcinoma (pRCC) (*Linehan & Ricketts, 2019*). CCRCC is a metabolic disease (*Wettersten et al., 2017*), accounting for more than 80% of all renal cell carcinomas (*Makhov et al., 2018*). It is the most aggressive subtype of renal cell carcinoma with a high rate of metastasis and recurrence (*Jiang et al., 2020*; *Yuan et al., 2018*). Although some progress has been made in the treatment of CCRCC, the current treatments of CCRCC still focus on surgical treatment and traditional chemotherapy (*Loo et al., 2019*; *Bex et al., 2019*). At the same time, there is a lack of effective early diagnosis methods in the clinic, and some patients still have a relapse and targeted drug tolerance, leading to poor prognosis of radiotherapy and chemotherapy. Therefore, finding new targeted biomarkers relevant to the diagnosis and treatment of CCRCC remains of paramount importance.

In this study, bioinformatics methods were used to obtain CCRCC gene expression data from GEO database, and normal samples and CCRCC samples were selected for grouping processing. Next, 738 DEGs were screened, including 289 up-regulated genes and 449 down-regulated genes. And then, GO enrichment analysis and KEGG signal pathway analysis were performed by DAVID. The up-regulated DEGs are mainly concentrated on protein binding, plasma membrane, inflammation, signal transduction, and PI3K-Akt signaling pathways, while the down-regulated genes are mainly concentrated on the extracellular exosome, oxidation–reduction process, integral component of membrane, protein homodimerization activity, and metabolic pathways. Finally, the top 35 hub genes were screened by PPI network. Based on the novelty of ITGAX, LAPTM5 and SERPINE1 that have not been reported in CCRCC, the expression of these three genes is significantly associated with survival prognosis and all have high degrees. Therefore, ITGAX, LAPTM5, and SERPINE1 were finally selected as key genes. Further analysis showed that ITGAX, LAPTM5, and SERPINE1 are highly expressed in CCRCC, which are significantly related to the survival prognosis of CCRCC. The methylation level of ITGAX, LAPTM5, SERPINE1 in CCRCC is reduced. Moreover ITGAX, LAPTM5 and SERPINE1 are related to the Clinical pathology of CCRCC and have good diagnostic value for CCRCC. In conclusion, we provided a systematic and comprehensive analysis of CCRCC and is the first to suggest that ITGAX, LAPTM5, and SERPINE1 might be used as biomarkers for the new clinical diagnosis and treatment of CCRCC.

## MATERIALS & METHODS

### Data collection

Four gene expression profiles ; (GSE12606, GSE53000, GSE68417, and GSE66272) were downloaded from the GEO (*Barrett et al., 2013*) database (http://www.ncbi.nlm. nih.gov/geo/). The TCGA KIRC data was downloaded from TCGA (https://www.cancer.gov/about-nci /organization /ccg/research/structural-genomics/tcga). The GSE12606 (*Stickel et al., 2009*) data set was obtained by GPL570 platform, including three adjacent normal kidney specimens, and three CCRCC samples. The GSE53000 (*Gerlinger et al., 2014*) and GSE68417 (*Thibodeau et al., 2016*) data sets were based on GPL6244 Platforms. GSE53000 data set was comprised of 60 samples including 6 adjacent normal kidney specimens and 54 CCRCC samples and GSE68417 data set was comprised of 43 samples including 14 adjacent normal kidney specimens and 29 CCRCC samples. GSE66272 (*Wotschofsky et al., 2016*; *Liep et al., 2016*) data set was obtained by GPL5029 platform and contained 27 CCRCC tumor samples and 26 normal kidney samples. The TCGA KIRC data contained 539 CCRCC tumor samples and 72 normal kidney samples.

### Study design and data processing

In order to clarify our study, we designed a flow chart to demonstrate data collection, processing, analysis and verification (Fig. 1). The online tool GEO2R (https://www.cancer.gov/about- nci/organization/ccg/research/structural-genomics/tcga) is an online analysis of GEO series based on the R programming language, which is used to screen DEGs between normal kidney and CCRCC samples from the GSE data sets. The data was standardized and filtered to select significant DEGs. $P$-value < 0.05, log Fold Change ($|$log FC$|$) $\geq$1 were chosen as the cutoff criteria. log Fold Change ($|$ log FC $|$) $\geq$ 1 means that the multiple of change is greater than or equal to 2. It is generally considered that there is a difference between 2 times and more. Then, DEGs were further screened according to cutoff criteria: $P$-value < 0.05, log FC $\geq$1 as up-regulated genes, $P$-value < 0.05, log FC $\leq$1 as down-regulated genes. Finally, importing all the up-regulated genes or down-regulated genes in the 4 datasets into Funrich 3.1.1 software, and taking the intersection of the up-regulated genes or down-regulated genes in the 4 datasets respectively. TCGA RNA-seq simple converter was used to standardize and log2 conversion of TCGA KIRC data. TCGA KIRC data were used for expression verification and ROC curve analysis of Key genes.

### Enrichment analysis of DEGs

DAVID 6.8 (Database for Annotation, Visualization and Integrated Discovery, https://david.ncifcrf.gov/) (*Huang da, Sherman & Lempicki, 2009b*; *Huang da, Sherman & Lempicki, 2009a*) database was used to analyze Gene Ontology (GO) (*The Gene Ontology Consortium, 2017*) such as the biological process (BP) (*Berchtold, Csaba & Zimmer, 2017*), cellular component (CC) (*Borg & Baudino, 2011*) andmolecular function (MF), and Kyoto Encyclopedia of Genes and Genomes (KEGG) (*Kanehisa & Goto, 2000*) pathway enrichment analysis of important DEGs, which promoted the visualization of gene and protein function (*Dennis Jr et al., 2003*). Among them, the count value represents the number of genes enriched in the pathway. The cutoff value was $P$ < 0.05.
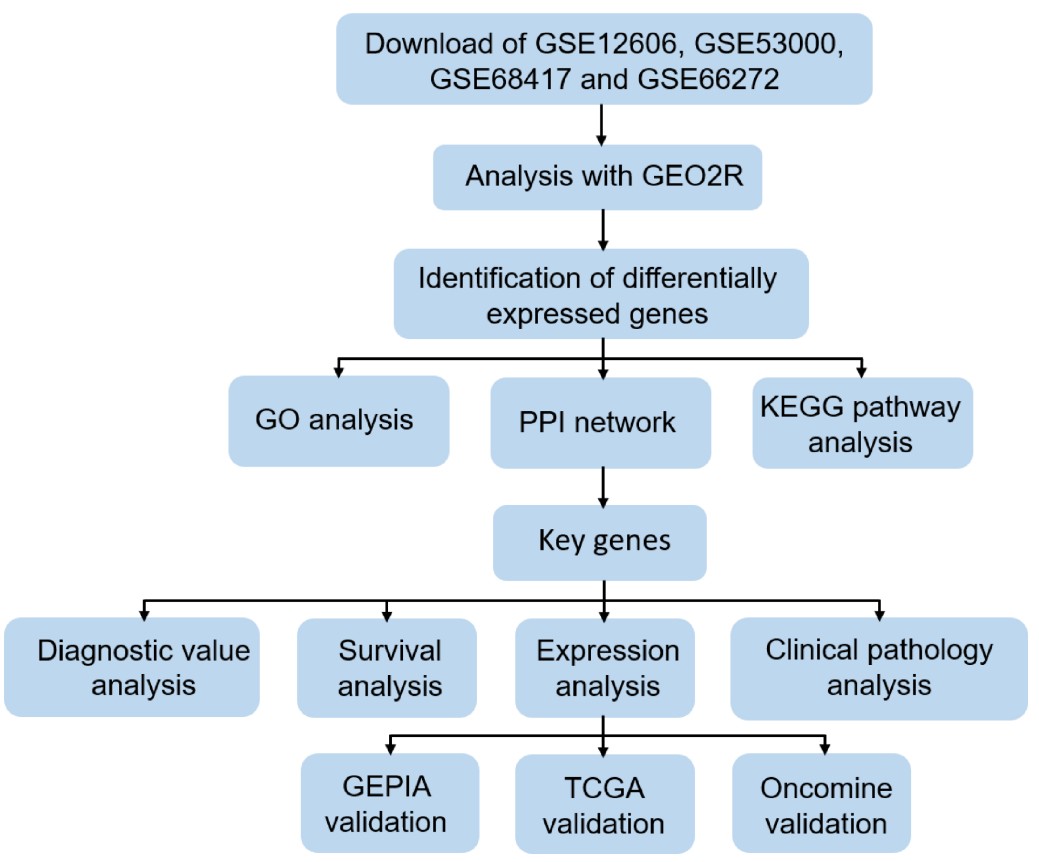

**Figure 1** Flow chart of data collection, processing, analysis and verification in this study.

## Protein-Protein Interaction (PPI) Network construction and hub gene screening

The online tool STRING (*Von Mering et al., 2003*) (http://string-db.org) was utilized for the analysis of protein-protein interactions . Importing all DEGs (including up-regulated genes and down-regulated genes) into the STRING database for analysis, and the confidence level $\geq 0.4$ is considered to be significant for PPI. STRING and Cytoscape software (version 3.6.1) were used to construct the PPI network, which is regarded as a network software platform for visualizing protein molecular interactions. Using the degree algorithm in the cytohubba plug-in of Cytoscape, the degree value could be calculated and thus the hub genes could also be screened from PPI network. The importance of genes is directly related to the degree of protein. According to the degree value (degree $\geq 37$), the first 35 genes were selected as hub genes. Three key genes ITGAX, LAPTM5 and SERPINE1 with high degree were screened through survival and prognosis analysis as well as the novelty of genes.

## Key genes analysis and verification

Three key genes were analyzed and verified comprehensively by using TCGA, GEPIA, Oncomine, Kaplan–Meier plotter and UALCAN databases. Similarly, TCGA database also applied to ROC curve analysis. ROC curve was plotted by R software, which could be used to

analyze the diagnostic value of key genes in CCRCC. GEPIA (http://gepia.cancer-pku.cn/) analyzed the expression of key genes in CCRCC. Oncomine (*Rhodes et al., 2007*) (http://www.on comine.org) was a database consisting of microarray data of various tumors, which verified the expression of key genes in CCRCC. Kaplan-Meier Plotter (https://kmplot.com/analysis/) was used to analyze the survival of key genes in CCRCC. UALCAN (*Chandrashekar et al., 2017*) (http://ualcan.path. uab.edu/index.html) online database was used to analyze the relationship between key genes and Clinical pathology in CCRCC, and the promoter methylation level of key genes in CCRCC was analyzed.

## Statistical analysis

Statistically significant differences between the normal tissues group and tumor tissues group were determined using Student's t tests. Also, Kaplan–Meier analysis was used to assess OS. ROC curve is a graphical plot (*Cao & López-de Ullibarri, 2019*) that reflects the sensitivity and specificity of continuous variables. $P < 0.05$ was considered to be statistically significant for all the tests. All the statistical analyses applied GraphPad prism 6.0 or R software.

# RESULTS

## Volcano plots of the differentially expressed genes in four datasets

The data sets of CCRCC, which were GSE12606, GSE53000, GSE68417 and GSE66272, were downloaded from the GEO database (Table 1) and analyzed by GEO2R separately. A total of 9,560 DEGs were screened from the GSE12606 data sets, among which 1568 genes were up-regulated and 5436 genes were down-regulated instead. A total of 1,286 DEGs were screened from the GSE53000 data sets, among which 583 were up-regulated and 708 were down-regulated. A total of 1,890 DEGs were screened from the GSE68417 data sets among them 726 up-regulated and 1,164 down-regulated genes were selected. There were 3,627 up-regulated genes and 5,170 down-regulated genes among the 8,797 DEGs screened from the GSE66272 data sets. The screening criteria were *P*-value 0.05, log Fold Change ($|\log FC|) \geq 1$. All of the DEGs from the four data sets were presented in Volcano plots (Figs. 2A–2D). Among them, red represents high-expressed genes, green represents low-expressed genes, and black represents genes whose expression levels are not significant in each data set. We gained the intersection of four independent data sets through Funrich 3.1.1 software and made a visualization analysis of Venn Diagram. After being overlapped, the common 738 genes (Fig. S1) were identified, including 289 up-regulated and 449 down-regulated genes (Figs. 2E–2F).

## GO and KEGG enrichment analysis of DEGs

To further explore the function of DEGs, enrichment analysis of up-regulated genes and down-regulated genes were displayed respectively. DAVID 6.8 was used to perform GO and KEGG analysis of DEGs in CCRCC (Table 2). In biological processes, up-regulated DEGs are mostly involved in cell adhesion, signal transduction, immune response, especially the regulation of inflammatory response (Fig. 3A); while down-regulated DEGs are mostly involved in oxidation–reduction process proteolysis, ion transmembrane transport,

**Table 1  Summary of TCGA and GEO Clear Cell Renal Cell Carcinoma datasets.**

| Author, year | Sample | GEO no. | Platform | Normal | Tumor | (Refs.) |
|---|---|---|---|---|---|---|
| Stickel JS et al, 2009 | CCRCC | GSE12606 | GPL570 | 3 | 3 | *Stickel et al. (2009)* |
| Gerlinger M et al, 2014 | CCRCC | GSE53000 | GPL6244 | 6 | 54 | *Gerlinger et al. (2014)* |
| Thibodeau BJ et al, 2016 | CCRCC | GSE68417 | GPL6244 | 29 | 14 | *Thibodeau et al. (2016)* |
| Wotschofsky Z et al, 2016 | CCRCC | GSE66272 | GPL5029 | 27 | 26 | *Wotschofsky et al. (2016)* |
| **TCGA** | CCRCC | – | – | 72 | 539 | |

**Notes.**

TCGA, The Cancer Genome Atlas; GEO, Gene Expression Omnibus.

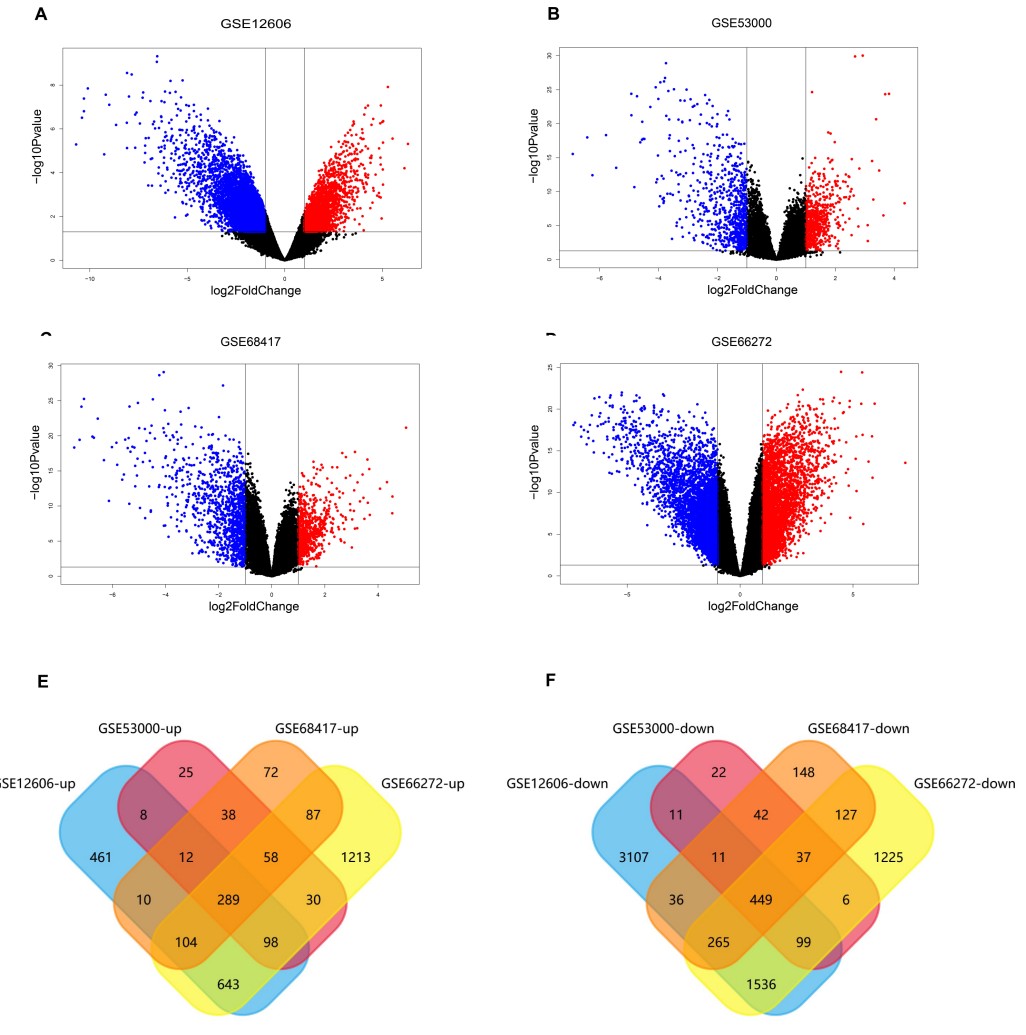

**Figure 2  Screening of differentially expressed genes.** (A–D) The volcano plot of all DEGs respectively in GSE12606, GSE53000, GSE68417, and GSE66272 datasets. Red and green nodes represent up-regulated genes and down-regulated genes, respectively. (E–F) 738 DEGs were identified in four profile datasets (GSE12606, GSE53000, GSE68417, and GSE66272), 289 upregulated genes, 449 downregulated genes.

response to drug, and ion transport (Fig. 3E). In terms of cellular components, up-regulated DEGs are mainly distributed in the plasma membrane, integral component of the membrane, extracellular exosome, and membrane (Fig. 3B); down-regulated DEGs are mainly distributed in integral component of membrane, plasma membrane, cellular exosomes, and integral component of plasma membrane (Fig. 3F). In terms of molecular function, up-regulated DEGs generally have protein binding, identical protein bind, ATP binding capacity, and protein homodimerization activity (Fig. 3C); down-regulated DEGs generally have identical protein homodimerization activity, calcium ion binding, sequence-specific DNA binding, and oxidoreductase activity (Fig. 3G). In the KEGG signal pathway, the up-regulation of DEGs mostly involved in the PI3K-Akt signaling pathway, pathways in cancer, focal adhesion, and HIF-1 signaling pathway (Fig. 3D); while the down-regulation of DEGs mainly involved in the metabolic pathway, Biosynthesis of antibiotics, Carbon metabolism, and Aldosterone-regulated sodium reabsorption (Fig. 3H). CCRCC is a kind of metabolic disease. Metabolic reprogramming covers different processes including aerobic glycolysis, fatty acid metabolism and the utilization of tryptophan, glutamine as well as arginine (*Lucarelli et al., 2019*), which has also been proved by the results of KEGG pathway enrichment analysis in the work. KEGG pathway results show that down-regulated DEGs are enriched in metabolic pathways, Glycolysis/Gluconeogenesis, Glycine, serine and threonine metabolism and so on.

In addition, GO and KEGG enrichment analyses were performed for all DEG (up-regulated and down-regulated genes) (Fig. S2). Finally, 11 common enrichment results were found from the three analyses (including up-regulated DEG, down-regulated DEG and all DEG enrichment analyses). The common enrichment results are as follows: response to drug, plasma membrane, integral component of membrane, extracellular space, extracellular exosome, cell surface, cell surface, protein homodimerization activity, identical protein binding. The results of enrichment analysis were consistent with previous studies (*Wang, Yu & Chai, 2019*; *Tian et al., 2019*). It suggests that these 11 pathways may be important in CCRCC.

## Construction of PPI network and screening of key genes

To identify the key genes, the STRING online database and Cytoscape software were used to analyze all DEGs (including up-regulated genes and down-regulated genes)and construct PPI network (Fig. 4A). Based on the main role of proteins in biological functions, their interaction determines the molecular and cellular mechanisms that control the health and disease state of the organism (*Safari-Alighiarloo et al., 2014*). Next, using the degree algorithm in the cytohubba plug-in of Cytoscape to screen the hub genes in PPI network. The gene whose degree ≥37 could be defined hub gene, therefore the first 35 genes (Table 3) in the PPI network were chosen to be hub genes (Fig. 4D). Next, GO enrichment analysis was conducted for all DEGs and the first 35 DEGs in the PPI network. The result showed that all DEGs mainly enriched in signal transduction, oxidation–reduction process, cell adhesion, inflammatory response, plasma membrane, integral component of membrane, extracellular exosome, integral component of plasma membrane (Figs. 4B–4C). The top 35 DEGs were enriched in extracellular space, extracellular exosome, inflammatory response,
**Table 2  GO and KEGG analysis of differentially expressed DEG associated with CCRCC.**

| Expression | Category | Term | Count | *P*-value |
|---|---|---|---|---|
| **Upregulated** | GO_BP | GO:0006954~inflammatory response | 30 | 1.30E−12 |
| | GO_BP | GO:0007165~signal transduction | 29 | 1.31E−02 |
| | GO_BP | GO:0007155~cell adhesion | 26 | 5.33E−08 |
| | GO_BP | GO:0006955~immune response | 24 | 1.76E−07 |
| | GO_BP | GO:0042493~response to drug | 21 | 5.92E−08 |
| | GO_BP | GO:0045087~innate immune response | 19 | 1.36E−04 |
| | GO_BP | GO:0030198~extracellular matrix organization | 18 | 1.08E−08 |
| | GO_BP | GO:0050900~leukocyte migration | 17 | 6.55E−11 |
| | GO_BP | GO:0001525~angiogenesis | 17 | 4.06E−07 |
| | GO_BP | GO:0008284~positive regulation of cell proliferation | 17 | 2.57E−03 |
| | GO_CC | GO:0005886~plasma membrane | 108 | 1.13E−09 |
| | GO_CC | GO:0016021~integral component of membrane | 107 | 2.03E−04 |
| | GO_CC | GO:0070062~extracellular exosome | 66 | 2.68E−04 |
| | GO_CC | GO:0016020~membrane | 64 | 4.11E−07 |
| | GO_CC | GO:0005887~integral component of plasma membrane | 62 | 6.07E−14 |
| | GO_CC | GO:0005576~extracellular region | 50 | 1.98E−06 |
| | GO_CC | GO:0005615~extracellular space | 49 | 2.48E−08 |
| | GO_CC | GO:0009986~cell surface | 31 | 1.08E−09 |
| | GO_CC | GO:0005783~endoplasmic reticulum | 20 | 4.86E−02 |
| | GO_CC | GO:0000139~Golgi membrane | 17 | 1.94E−02 |
| | GO_MF | GO:0005515~protein binding | 154 | 6.05E−03 |
| | GO_MF | GO:0005524~ATP binding | 36 | 6.15E−03 |
| | GO_MF | GO:0042803~protein homodimerization activity | 21 | 7.66E−03 |
| | GO_MF | GO:0042802~identical protein binding | 20 | 1.92E−02 |
| | GO_MF | GO:0005102~receptor binding | 19 | 7.41E−06 |
| | GO_MF | GO:0004872~receptor activity | 14 | 2.70E−05 |
| | GO_MF | GO:0004672~protein kinase activity | 12 | 2.09E−02 |
| | GO_MF | GO:0005215~transporter activity | 8 | 3.41E−02 |
| | GO_MF | GO:0004888~transmembrane signaling receptor activity | 8 | 4.43E−02 |
| | GO_MF | GO:0005201~extracellular matrix structural constituent | 7 | 5.17E−04 |
| | KEGG | hsa04151: PI3K-Akt signaling pathway | 19 | 2.01E−04 |
| | KEGG | hsa04510: Focal adhesion | 17 | 3.76E−06 |
| | KEGG | hsa05200: Pathways in cancer | 17 | 5.96E−03 |
| | KEGG | hsa04066: HIF-1 signaling pathway | 12 | 3.37E−06 |
| | KEGG | hsa04015: Rap1 signaling pathway | 12 | 3.56E−03 |
| | KEGG | hsa04145: Phagosome | 10 | 3.39E−03 |
| | KEGG | hsa05205: Proteoglycans in cancer | 10 | 2.05E−02 |
| | KEGG | hsa05150: Staphylococcus aureus infection | 9 | 1.14E−05 |
| | KEGG | hsa04611: Platelet activation | 9 | 4.86E−03 |
| | KEGG | hsa05133: Pertussis | 8 | 7.79E−04 |

*(continued on next page)*

**Table 2** (*continued*)

| Expression | Category | Term | Count | *P*-value |
|---|---|---|---|---|
| **Downregulated** | GO_BP | GO:0055114~oxidation–reduction process | 33 | 3.33E−05 |
| | GO_BP | GO:0034220~ion transmembrane transport | 19 | 5.29E−06 |
| | GO_BP | GO:0042493~response to drug | 18 | 1.62E−03 |
| | GO_BP | GO:0006811~ion transport | 14 | 1.66E−05 |
| | GO_BP | GO:0055085~transmembrane transport | 14 | 8.16E−03 |
| | GO_BP | GO:0007588~excretion | 13 | 4.97E−11 |
| | GO_BP | GO:0010628~positive regulation of gene expression | 13 | 3.13E−02 |
| | GO_BP | GO:0006814~sodium ion transport | 12 | 4.94E−06 |
| | GO_BP | GO:0001822~kidney development | 12 | 8.92E−06 |
| | GO_BP | GO:0008152~metabolic process | 12 | 3.22E−03 |
| | GO_CC | GO:0016021~integral component of membrane | 171 | 4.63E−07 |
| | GO_CC | GO:0070062~extracellular exosome | 170 | 1.01E−33 |
| | GO_CC | GO:0005886~plasma membrane | 162 | 2.99E−12 |
| | GO_CC | GO:0005887~integral component of plasma membrane | 76 | 3.16E−11 |
| | GO_CC | GO:0005615~extracellular space | 50 | 1.87E−03 |
| | GO_CC | GO:0016324~apical plasma membrane | 44 | 4.39E−22 |
| | GO_CC | GO:0016323~basolateral plasma membrane | 28 | 2.05E−14 |
| | GO_CC | GO:0009986~cell surface | 24 | 5.53E−03 |
| | GO_CC | GO:0005759~mitochondrial matrix | 19 | 8.74E−04 |
| | GO_CC | GO:0043025~neuronal cell body | 15 | 1.88E−02 |
| | GO_MF | GO:0042803~protein homodimerization activity | 33 | 7.43E−04 |
| | GO_MF | GO:0005509~calcium ion binding | 29 | 7.77E−03 |
| | GO_MF | GO:0043565~sequence-specific DNA binding | 20 | 4.43E−02 |
| | GO_MF | GO:0016491~oxidoreductase activity | 13 | 3.24E−03 |
| | GO_MF | GO:0046983~protein dimerization activity | 11 | 3.31E−03 |
| | GO_MF | GO:0008201~heparin binding | 11 | 5.22E−03 |
| | GO_MF | GO:0005088~Ras guanyl-nucleotide exchange factor activity | 10 | 1.78E−03 |
| | GO_MF | GO:0003824~catalytic activity | 10 | 3.74E−02 |
| | GO_MF | GO:0016787~hydrolase activity | 10 | 4.07E−02 |
| | GO_MF | GO:0046934~phosphatidylinositol-4,5-bisphosphate 3-kinase activity | 9 | 1.10E−04 |
| | KEGG | hsa01100: Metabolic pathways | 68 | 8.56E−08 |
| | KEGG | hsa01130: Biosynthesis of antibiotics | 19 | 5.17E−05 |
| | KEGG | hsa01200: Carbon metabolism | 11 | 1.83E−03 |
| | KEGG | hsa04960: Aldosterone-regulated sodium reabsorption | 9 | 1.45E−05 |
| | KEGG | hsa04978: Mineral absorption | 9 | 3.69E−05 |
| | KEGG | hsa00010: Glycolysis / Gluconeogenesis | 9 | 7.57E−04 |
| | KEGG | hsa04610: Complement and coagulation cascades | 8 | 4.14E−03 |
| | KEGG | hsa04966: Collecting duct acid secretion | 7 | 1.11E−04 |
| | KEGG | hsa00260: Glycine, serine and threonine metabolism | 7 | 9.10E−04 |
| | KEGG | hsa00280: Valine, leucine and isoleucine degradation | 7 | 2.46E−03 |

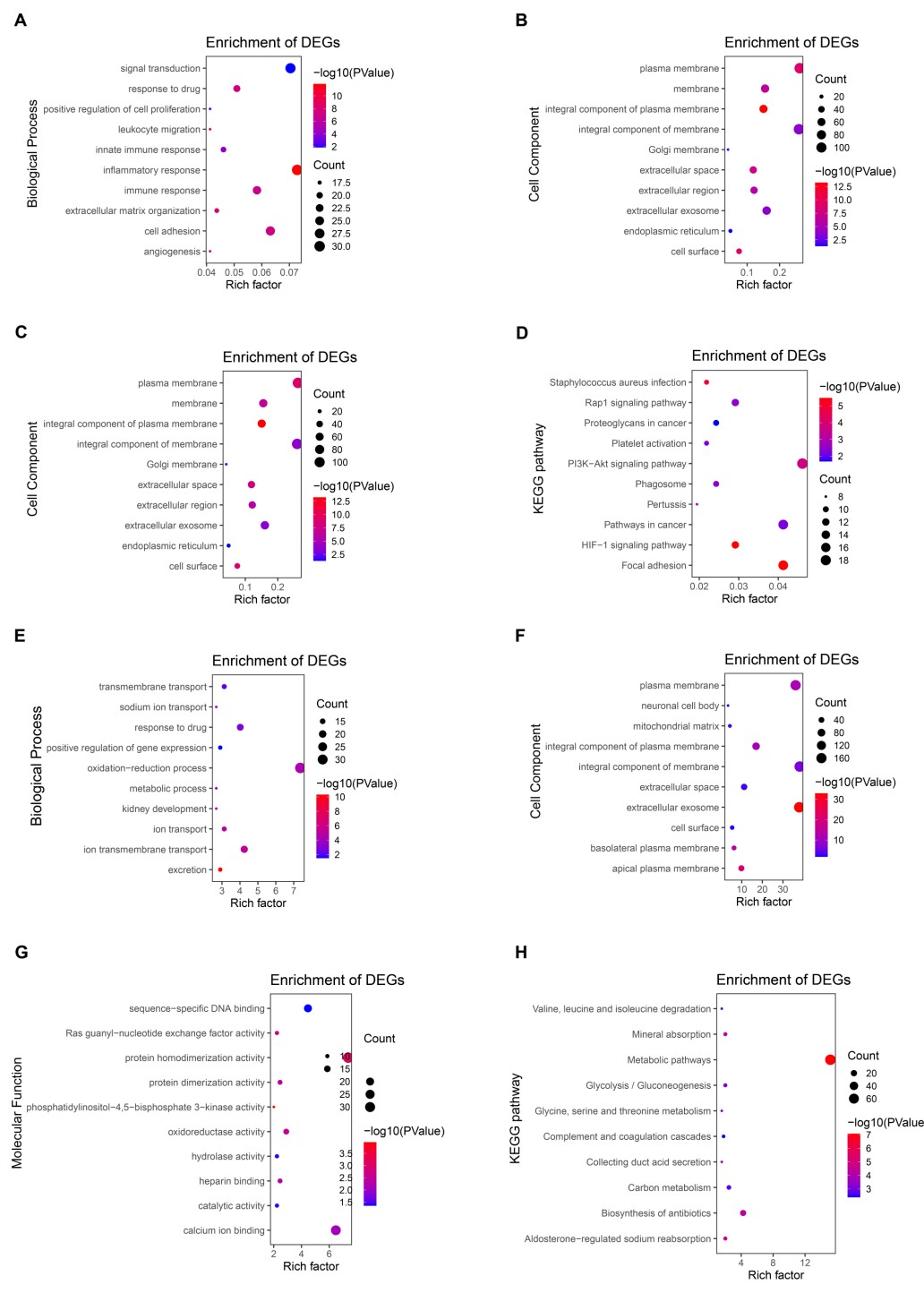

**Figure 3** **Enrichment analysis of GO and KEGG with up-regulated DEG.** (A) The biological process of GO analysis showed that the up-regulation of DEGs was mainly related to cell adhesion, inflammatory response, signal transduction, and immune response. (B) The enrichment analysis of up-regulated DEGs cell components is mainly related to the cellular exosomes, integral component of membrane, plasma membrane, and integral component of plasma membrane. 

**Figure 3 (...continued)**
(C) The molecular function of GO analysis showed that the up-regulation of DEGs was mainly related to protein binding, identical protein bind, ATP binding capacity, and protein homodimerization activity. (D) The KEGG pathways related to the up-regulation of DEGs expression mainly include the PI3K-Akt signaling pathway, focal adhesion, pathways in cancer, and HIF-1 signaling pathway. (E) The biological process of GO analysis showed that the downregulation of DEGs was mainly related to oxidation–reduction process proteolysis, ion transmembrane transport, response to the drug, and ion transport. (F) The enrichment analysis of down-regulated DEGs cell components is mainly related to integral components of membrane, plasma membrane, cellular exosomes, and integral component of plasma membrane. (G) The molecular function of GO analysis showed that the downregulation of DEGs was mainly related to protein homodimerization activity, calcium ion binding, oxidoreductase activity, and sequence-specific DNA binding. (H) The KEGG pathways related to the down-regulation of DEGs expression mainly include metabolic pathway, Biosynthesis of antibiotics, Carbon metabolism, and Aldosterone-regulated sodium reabsorption.

**Table 3** The top 35 DEGs identified with the degree in the PPI network (Degree ≥ 37).

| Expression | Genes | Degree | Genes | Degree |
|---|---|---|---|---|
| **Upregulated** | EGFR | 110 | C3AR1 | 52 |
| | VEGFA | 100 | CYBB | 51 |
| | PTPRC | 93 | LCP2 | 50 |
| | FN1 | 90 | TLR7 | 49 |
| | ITGB2 | 70 | C3 | 45 |
| | TLR2 | 67 | CCL5 | 44 |
| | MMP9 | 66 | TIMP1 | 43 |
| | CD86 | 64 | HCK | 43 |
| | CXCR4 | 62 | FCGR2A | 42 |
| | ICAM1 | 62 | CD53 | 41 |
| | CCND1 | 61 | CTSS | 41 |
| | TYROBP | 59 | LAPTM5 | 40 |
| | CSF1R | 58 | LOX | 38 |
| | PLEK | 55 | SERPINE1 | 37 |
| | ITGAX | 53 | CAV1 | 37 |
| **Downregulated** | ALB | 134 | KNG1 | 52 |
| | EGF | 80 | KIT | 44 |
| | ERBB2 | 67 | | |

an integral component of the plasma membrane, and cell surface (Fig. 4E). By comparison, it turned out that the enrichment analysis results of all DEGs in PPI network contained the results of the top 35's. In the end, three new key genes with high degree which were ITGAX, LAPTM5 and SERPINE1, could be screened by using GEPIA and Kaplan–Meier plotter database.

## The expression of key genes

Among the 35 genes, we focused on ITGAX, LAPTM5, and SERPINE1, which have not been reported to be related to the occurrence and development of CCRCC. Firstly,

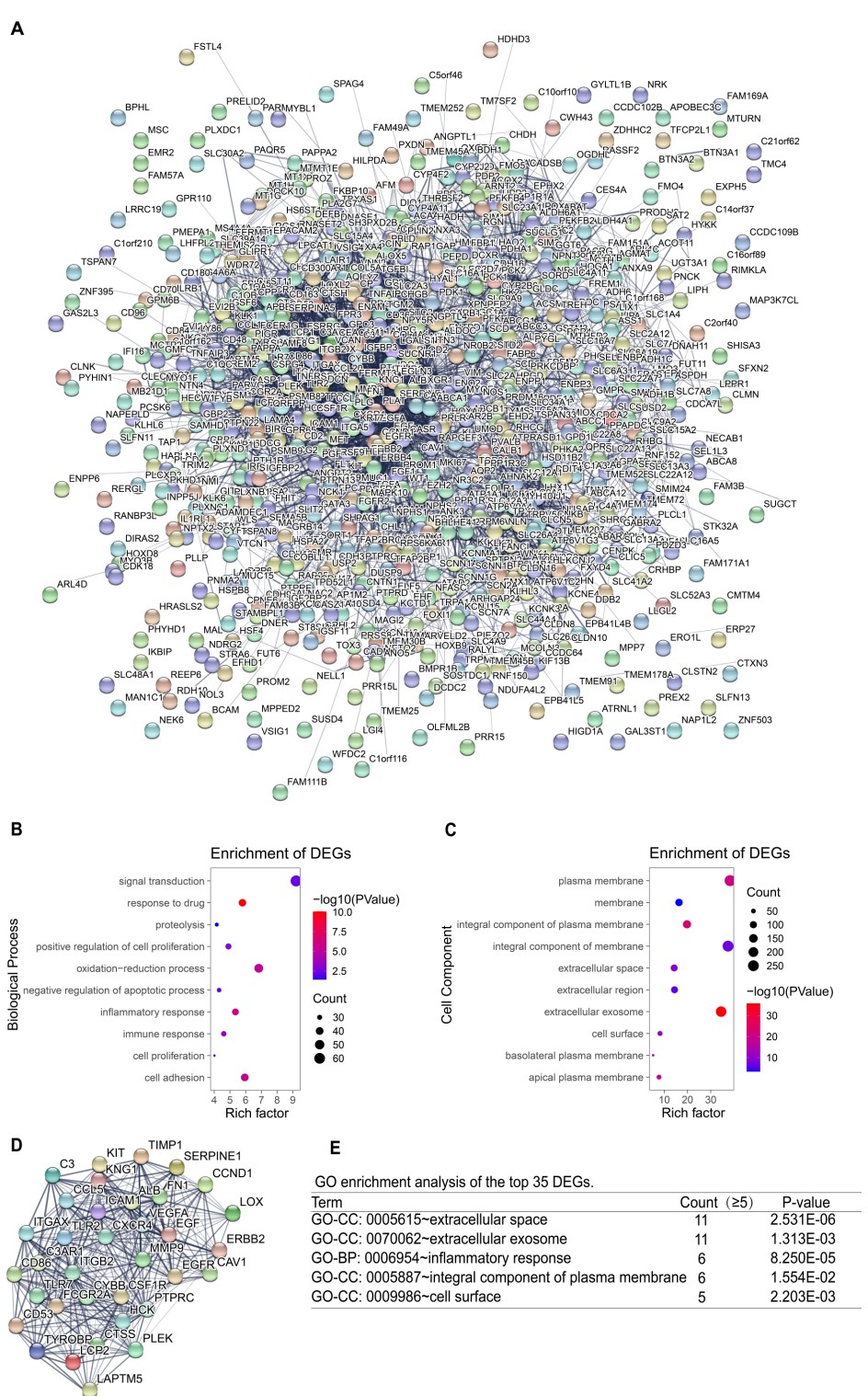

**Figure 4** **PPI network and GO enrichment analysis of key genes.** (A) The PPI network is constructed, including up-regulated genes and down-regulated genes, and considers that the confidence level ≥ 0.4 is significant for PPI. (B–C) GO enrichment analysis of all DEGs in PPI network. (D) PPI network of the top 35 DEGs according to the degree (degree ≥ 37). (E) GO enrichment analysis of the top 35 DEGs.

the expression levels of ITGAX, LAPTM5, and SERPINE1 in CCRCC tumor tissues are significantly higher than those in normal tissues adjacent to cancer according to GEPIA database (Figs. 5A–5C). Furthermore, the data of TCGA KIRC showed that compared with adjacent normal tissues, the mRNA expressions of ITGAX, LAPTM5, and SERPINE1 in 72 pairs of CCRCC tissues were significantly increased (Figs. 5D–5F). In the Oncomine Gumz renal database, the mRNA levels of ITGAX, LAPTM5 and SERPINE1 were also upregulated in CCRCC tissues when compared with adjacent normal kidney tissues (Figs. 5H–5G). According to the UALCAN database, the promoter methylation levels of ITGAX, LAPTM5 and SERPINE were decreased in CCRCC (Figs. 5J–5L). To sum up, according to the GEPIA database, Oncomine Gumz renal database and TCGA database, the expression levels of these three genes in CCRCC tumor tissues are significantly higher than those in normal tissues adjacent to cancer. It could be further speculated that the high expression of ITGAX, LAPTM5, and SERPINE1 in CCRCC tumor tissue might be related to the decrease of promoter methylation.

## The association of key genes expression with clinical pathology in CCRCC

Furthermore, the relationship between the mRNA expression of ITGAX, LAPTM5 and SERPINE1 and different clinical pathology grades were measured. The results showed that their mRNA expression was significantly related to pathological grades (Figs. 6A–6C). And the expression of ITGAX, LAPTM5, and SERPINE1 mRNA in CCRCC samples are also significantly correlated with severe clinical staging (Figs. 6D–6F). Among them, the expression levels of ITGAX, LAPTM5, and SERPINE1 were higher in stage 4 and grade 4. In conclusion, ITGAX, LAPTM5, and SERPINE1 are significantly associated with clinical pathology.

## Survival and diagnostic value of ITGAX, LAPTM5, and SERPINE1 in CCRCC

According to the Kaplan–Meier plotter database, the overall survival of ITGAX, LAPTM5, and SERPINE1 genes was tested (Figs. 7A–7C). The results showed that the high expression of three key genes in CCRCC was negatively correlated with prognosis. Then, the ROC curve was used to evaluate the difference between CCRCC and the normal tissues in the TCGA KIRC data. The ROC curves (Figs. 7D–7E) of these three genes showed that their area under the curve (AUC) and the 95% confidence intervals (CI) are as follows: ITGAX (AUC = 96.315, CI = 0.9423–0.9840), LAPTM5, (AUC = 94.808 CI = 0.9243–0.9719), and SERPINE1(AUC = 76.272, CI [0.7068–0.8186]). Since ROC curves had good specificity and sensitivity, ITGAX, LAPTM5, and SERPINE1 had excellent diagnostic efficiency for distinguishing tumors and normal tissues.

## DISCUSSION

Clear Cell Renal Cell Carcinoma (CCRCC) is a metabolic disease whose morbidity is rising worldwide. The feature of kidney cancer is to participate in the target genes' mutation of metabolic pathways (*Lucarelli et al., 2019*). Recently, with the application of bioinformatics,

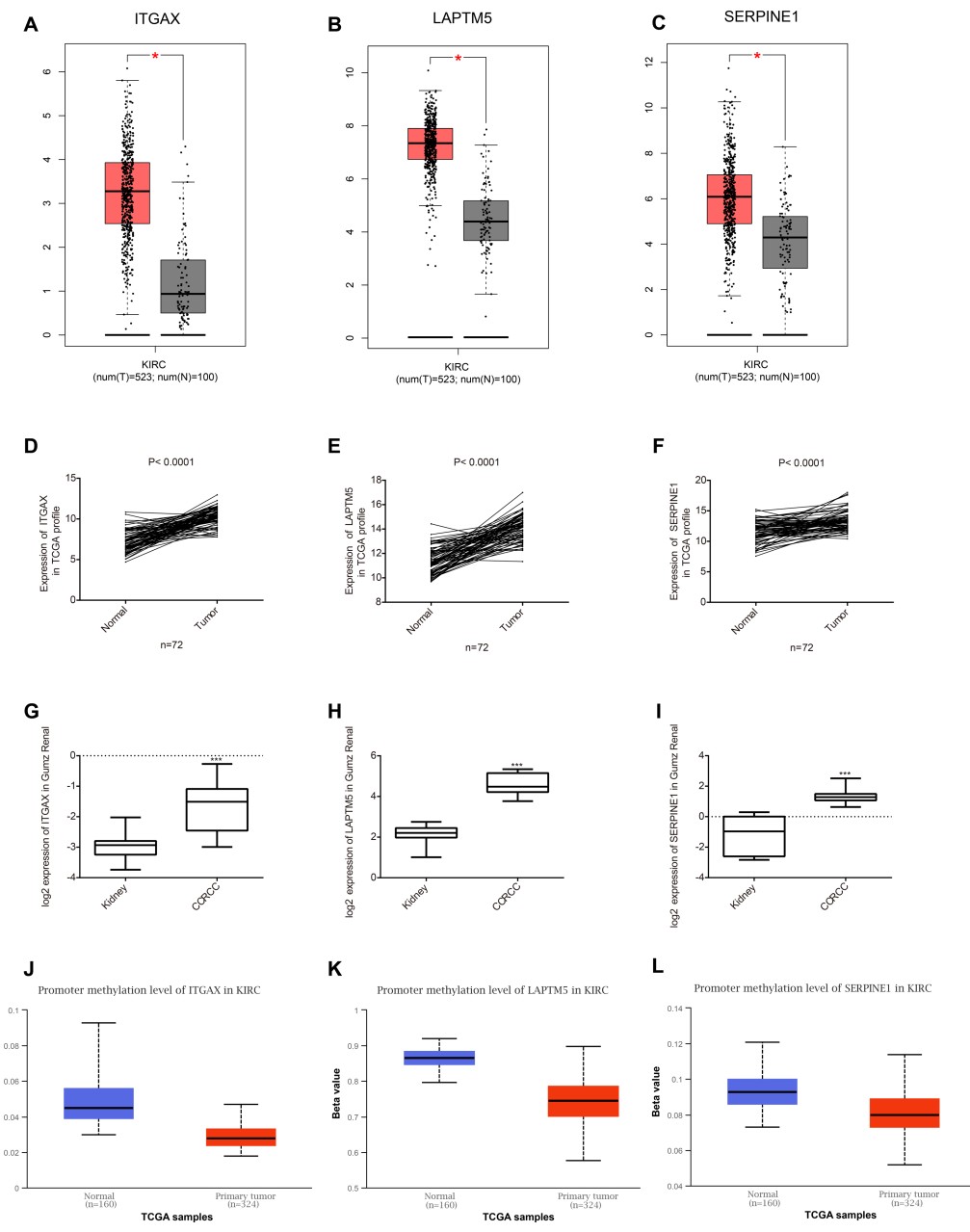

**Figure 5  Key gene expression between normal kidney and CCRCC tissues.** (A–C) Box plot showing the expression of ITGAX, LAPTM5, SERPINE1 in GEPIA database. These three genes are highly expressed in CCRCC. (D–F) In the TCGA database, compared with adjacent normal tissues, the expression of ITGAX, LAPTM5, SERPINE1 mRNA in 72 pairs of CCRCC tissues increased significantly. (G–I) The expression of ITGAX, LAPTM5, SERPINE1 are significantly increased in Oncomine Gumz renal database. $*$ $P < 0.05$, $**$ $P < 0.01$, $***$ $P < 0.001$. (J–L) The box plot showing the promoter methylation levels of ITGAX, LAPTM5 and SERPINE1 in the UALCAN database. $P < 0.01$.

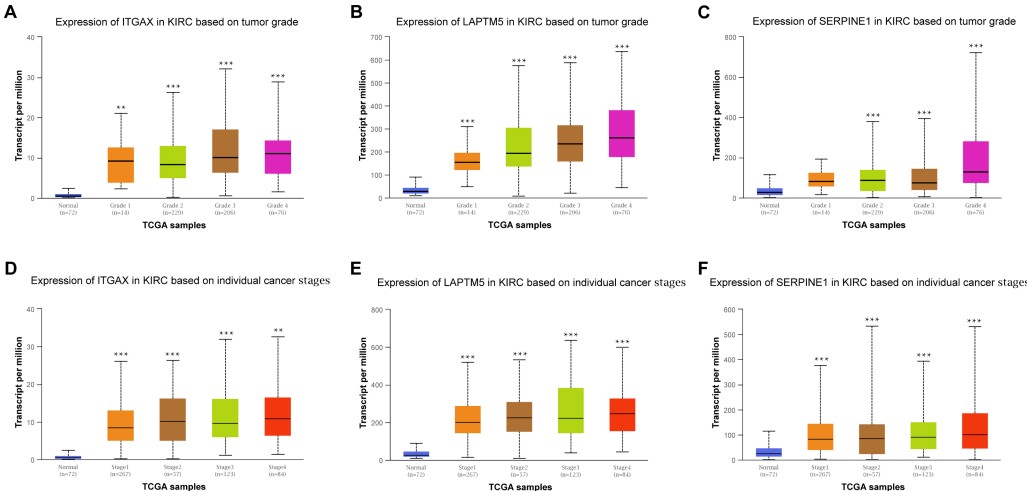

**Figure 6** **The relationship between ITGAX, LAPTM5, and SERPINE1 expression and clinical pathology of CCRCC.** (A–C) Boxplot showing that ITGAX, LAPTM5, and SERPINE1 mRNA expression were significantly related to pathological grades, and patients in grade 4 have the highest expression in CCRCC according to UALCAN databases. (D–F) Boxplot showing that the expression of ITGAX, LAPTM5, and SERPINE1 mRNA in CCRCC samples are significantly correlated with severe clinical staging and the mRNA expression of ITGAX, LAPTM5, and SERPINE1 were higher in patients with stage 4 according to UALCAN databases. * $P < 0.05$, ** $P < 0.01$, *** $P < 0.001$.

the molecular characteristics of CCRCC have been greatly improved and the development of targeted therapy has been promoted. These advances have significantly improved the median survival of patients with advanced disease. However, about 30% of CCRCC local patients will still relapse or metastasize after surgical removal of the tumor (*Li et al., 2019*). Around 1/3 of the metastatic patients had poor prognosis and rare high drug resistance rate. Under the circumstance of different treatment, the identification of biomarkers was needed urgently so as to predict the drug's effects (*Deleuze et al., 2020*). Therefore, identification of CCRCC key genes and prognostic judgment is still very crucial. In our research, some reported genes related to CCRCC, such as VEGFA (*Zeng et al., 2016*), EGFR (*Cossu-Rocca et al., 2016*), were also screened out. Vascular Endothelial Growth Factor A (VEGFA) is a member of the PDGF/VEGF growth factor family. VEGFA has a potential role in the diagnosis and treatment of CCRCC. VEGFA can inhibit the proliferation of CCRCC 786-O cells, promote cell apoptosis, and inhibit cell migration and invasion (*Zeng et al., 2016*). Epidermal growth factor receptor (EGFR) is closely related to the progression of many epithelial malignancies and is an important therapeutic target (*Cossu-Rocca et al., 2016*). EGFR is a cell surface protein, belonging to the ERBB family. EGFR binds to epidermal growth factor to induce receptor dimerization and tyrosine self phosphorylation, which eventually leads to cell proliferation (*Mitsudomi & Yatabe, 2010*). EGFR can activate a variety of signaling pathways, mainly MAPK / ERK and PI3K / AKT pathways (*Yarden & Sliwkowski, 2001*). Compared with the former 4 studies, this study found three new key genes which were ITGAX, LAPTM5 and SERPINE1, and applied several methods to do functional analysis and systematic research to key genes such as methylation level analysis,

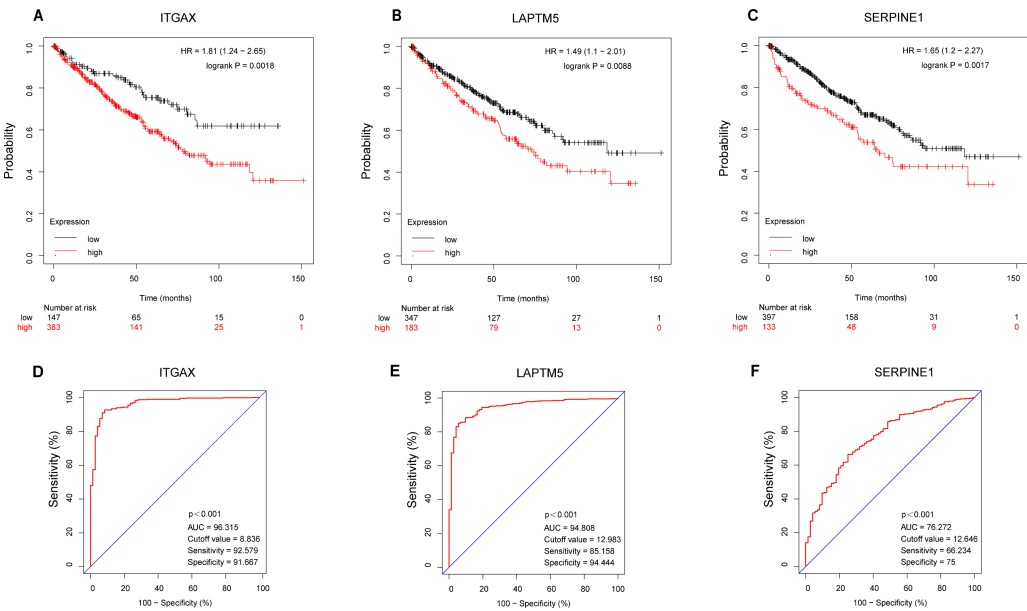

**Figure 7** **Survival and diagnostic value of ITGAX, LAPTM5, and SERPINE1 in CCRCC.** (A–C) The overall survival of ITGAX, LAPTM5, and SERPINE1.The results showed that the high expression of three key genes in CCRCC was negatively correlated with prognosis. (D) ROC curve of ITGAX (AUC = 96.315, cutoff value = 8.836, Sensitivity = 92.579 Specificity = 91.667). (E) ROC curve of LAPTM5, (AUC = 94.808, cutoff value = 12.983 Sensitivity = 85.158 Specificity = 94.444). (F) ROC curve of SERPINE1, AUC = 76.272, cutoff value = 12,646, Sensitivity = 66.234 Specificity = 75).

survival analysis, ROC curve analysis and so on. In previous studies, GSE12606 mainly focused on the functional analysis of HLA ligand in CCRCC, while GSE5300 mainly focused on the structure and revolution of CCRCC genome, and GSE66272 mainly focused on the role of miRNA in CCRCC and GSE68417 mainly focused on the analysis to the gene expression profile of CCRCC. On the contrary, this study put emphasis on screening the biomarkers used to do early diagnosis in CCRCC and analyzing the intersection of DEGs in four datasets.

Besides, three new potential marker genes ITGAX, LAPTM5, SERPINE1 were also screened out and proved. Integrin alpha x (ITGAX) is a member of the integrin family, commonly function as a receptor for extracellular matrix. It is reported that ITGAX is involved in the angiogenesis of dendritic cells and tumor angiogenesis (*Wang et al., 2019*). In addition, ITGAX is identified as a new type of aggressive prostate cancer susceptibility gene (*Williams et al., 2014*). Lysosomal protein transmembrane 5 (LAPTM5), known as E3 protein, may play a role in hematopoiesis and prevent excessive activation of lymphocytes (*Cai et al., 2015*). It is reported that LAPTM5 can regulate the proliferation and viability of bladder cancer cells, leading to cell cycle arrest in the G0/G1 phase (*Chen et al., 2017*). LAPTM5 is also associated with the spontaneous regression of neuroblastoma (*Inoue et al., 2009*). Studies have shown that Serpin family E member 1 (SERPINE1 is a regulator of Glioblastoma (GBM) cell proliferation, and is related to poor prognosis of patients and mesenchymal GBM. Down-regulation of SERPINE1 in primary

GBM cells inhibited the growth and invasiveness of tumors in the brain, and SERPINE1 plays a key role in the spread of GBM (*Seker et al., 2019*).

## CONCLUSIONS

To sum up, the expression levels of ITGAX, LAPTM5, SERPINE1 in CCRCC tumor tissues are significantly higher than those in normal tissues adjacent to cancer and are related to the tumor stage and tumor grade. ITGAX, LAPTM5, and SERPINE1 have high diagnostic efficiency for tumors and normal tissues, and their expressions are associated with poor prognosis of CCRCC. The decrease of promoter methylation of ITGAX, LAPTM5 and SERPINE1 in CCRCC tumor tissues indicates that the high expression of key genes in CCRCC might be relevant to the low methylation level. The limitation of this study lied in that the internal molecular mechanisms where key genes played a role remained unclear, which need a further research. Further studies are needed to explore the detailed mechanisms of these key genes in CCRCC. In conclusion, we identified ITGAX, LAPTM5, and SERPINE1 as potential marker genes of CCRCC by bioinformatics methods, providing insights for future therapeutic design. Meanwhile, we conducted a relatively systematic and comprehensive analysis on CCRCC data, thereby providing a theoretical basis for identifying therapeutic targets of CCRCC, promoting early detection, and monitoring tumor progression.

### Funding

This work was supported by grants from the National Natural Science Foundation of China [81702743] and China Postdoctoral Science Foundation [2018M640612, 2019T120568]. The funders had no role in study design, data collection and analysis, decision to publish, or preparation of the manuscript.

### Grant Disclosures

The following grant information was disclosed by the authors:
The National Natural Science Foundation of China: [81702743].
China Postdoctoral Science Foundation: [2018M640612, 2019T120568].

### Competing Interests

The authors declare there are no competing interests.

### Author Contributions

- Yingli Sui conceived and designed the experiments, performed the experiments, analyzed the data, prepared figures and/or tables, authored or reviewed drafts of the paper, and approved the final draft.
- Kun Lu and Lin Fu conceived and designed the experiments, analyzed the data, authored or reviewed drafts of the paper, and approved the final draft.

## Data Availability

The raw data and code are available in the Supplemental Files.

## Supplemental Information

Supplemental information for this article can be found online at http://dx.doi.org/10.7717/peerj.11272#supplemental-information.

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
