# Peer review of "Prediction and analysis of novel key genes ITGAX, LAPTM5, SERPINE1 in clear cell renal cell carcinoma through bioinformatics analysis"

_PeerJ, doi:10.7717/peerj.11272_

## Round 0.1 · original submission · Major Revisions

In this manuscript, the authors demonstrated three genes as the novel to predict the clear cell renal cell carcinoma (CCRCC) from the preexisting database. It is reasonable to select the overlapped DEGs in four different databases. However, several questions came up on the further analyses.

Could the authors run GO enrichment analysis using all genes (n = 289 + 449) regardless of its up- or down-regulating status and compared with two analyses shown in Figures 2 and 3? Maybe authors can find commonly enriched pathways from three analyses.

In Figure 4, please add the GO enrichment analysis of 4A and compare the enriched pathways between 4A and 4B. Also, describe or define “degree” in detail - How these numbers are calculated, if possible.

Please provide more underlying rationale on why three genes out of 35 are chosen, if possible. According to the degree, three gene are not in top 3.

The authors need to emphasize the novelty of the current work more by comparing four previous studies, especially if they provided GO enrichment analyses (e.g., Which pathways are newly found as the enriched in the current analysis in comparison to the previous 4 studies).

Reviewer 1 ·

Basic reporting

- The English language should be improved in Lines 20-24 to ensure your points are clear.
- Line 20-21: It is not explained why ROC curve is used for.
- Line 41: position of Reference 1 is not correct.
- Line 61-62: It is not clear how three key genes were selected out of 35 hub genes. This is important to clarify the criteria for feature selection.
- Data processing section, Line 79-87, needs more detailed description and English language needs to be improved. It is highly recommended to add a schematic/flowchart for this section so that the pipeline is clear and easy to understand.
- Grammar issue in Line 89: “… were used to analyze…”
- Scientific reporting language is not used in Line 94-102 and 110-114.
- Line 110: what are the two groups?

Experimental design

- A key missing piece of information is how three genes were isolated from the pool of top 35 differentially expressed genes. This is significantly important to be addressed in this work to understand the motivation and whether that can justify the feature selection part of the work.

Validity of the findings

- The findings are expected to be explored and discussed in much more depth rather than only reporting the observations. For example, in Line 131-149, only the differentially expressed genes and their GO are reported without discussing whether these observations have been reported in the literature.

Additional comments

- This article aims to find genes with high predictive power in CCRCC patients. This is an important study, however, there are major issues that need to be addressed.
- The results (Line 116-187) need to be explained in more details, characterized in more depth and discussed further.

Reviewer 2 ·

Basic reporting

The authors performed bioinformatics analysis to study the key genes in clear cell renal cell carcinoma. After different analyses, they convinced the roles of ITGAX, LAPTM5, SERPINE1 in CCRCC. Although the idea and objective are reasonable, there are some major points that need to be addressed:

- There are many grammatical errors and typos in this manuscript. The authors should go through to re-check and revise carefully.

- Another concern is that the authors missed mentioning a lot of literature review that focused on the same problem (bioinformatics analysis in CCRCC). At least they need to show what are the current processes on this specific problem and what is the novelty in this study.

Experimental design

- Source codes should be released for replicating the results.

- Did the authors concern about any batch effect removal when they merged all the datasets and performed analyses?

- Data is not much to convince the significance of the findings.

- ROC curve and AUC have been used in previous bioinformatics studies such as PMID: 32613242 and PMID: 31362508. Thus the authors are suggested to refer to more references in this description.

Validity of the findings

- Are the finding genes different from all previous works on the same problem?

- Table 2 is unreasonable, it is better if the authors display it as a heatmap figure.

- The authors should compare the performance with previous works on the same dataset.

Additional comments

no comment

Reviewer 3 ·

Basic reporting

no comments

Experimental design

no comments

Validity of the findings

no comments

Additional comments

1. Grammatical errors need to be rectified; some sentences need to be rephrased for easy understanding. Dangling sentences in the Methods section of the abstract. Sentences should not start with “AND.” Proofreading will help.
2. Can log2 change and multiplicity adjustments be considered?
3. The authors can explain briefly how the fold change cut-offs were chosen.
4. Comparison with previous work, limitations of the work, and future guidelines may be added.

Please address the above.

---

## Round 0.2 · accepted · Accept

The manuscript has been actively revised according to the reviewers' comments. I appreciate all your effort to address the answers. We are pleased to accept your paper in its current form.

Reviewer 3 ·

Basic reporting

NA

Experimental design

NA

Validity of the findings

NA

Additional comments

The authors have addressed the comments.